# Protecting brains and saving futures guidelines: A prospective, multicenter, and observational study on the use of telemedicine for neonatal neurocritical care in Brazil

**Gabriel Fernando Todeschi Variane**[1,2,3]*, **Maurício Magalhães**[1,3,4,5], **Rafaela Fabri Rodrigues Pietrobom**[1,3], **Alexandre Netto**[1,3], **Daniela Pereira Rodrigues**[3,6], **Renato Gasperini**[1,3,4], **Guilherme Mendes Sant'Anna**[7]

1 Neonatal Division, Department of Pediatrics, Irmandade da Santa Casa de Misericórdia de São Paulo, São Paulo, Brazil, 2 Neonatal Division, Grupo Santa Joana, São Paulo, Brazil, 3 Protecting Brains and Saving Futures Organization, São Paulo, Brazil, 4 Neonatal Unit, Hospital Israelita Albert Einstein, São Paulo, Brazil, 5 Faculdade de Ciências Médicas da Santa Casa de São Paulo, São Paulo, Brazil, 6 Pediatric Nursing Department, Escola Paulista de Enfermagem, Universidade Federal de São Paulo, São Paulo, Brazil, 7 Neonatal Division, McGill University Health Centre, Quebec, Canada

* gabriel.variane@pbsf.com.br

## Abstract

### Background

Management of high-risk newborns should involve the use of standardized protocols and training, continuous and specialized brain monitoring with electroencephalography (EEG), amplitude integrated EEG, Near Infrared Spectroscopy, and neuroimaging. Brazil is a large country with disparities in health care assessment and some neonatal intensive care units (NICUs) are not well structured with trained personnel able to provide adequate neurocritical care. To reduce this existing gap, an advanced telemedicine model of neurocritical care called *Protecting Brains and Saving Futures (PBSF) Guidelines* was developed and implemented in a group of Brazilian NICUs.

### Methods

A prospective, multicenter, and observational study will be conducted in all 20 Brazilian NICUs using the *PBSF Guidelines* as standard-of-care. All infants treated accordingly to the guidelines during Dec 2021 to Nov 2024 will be eligible. Ethical approval was obtained from participating centers. The primary objective is to describe adherence to the *PBSF Guidelines* and clinical outcomes, by center and over a 3-year period. Adherence will be measured by quantification of neuromonitoring, neuroimaging exams, sub-specialties consultation, and clinical case discussions and videoconference meetings. Clinical outcomes of interest are detection of seizures during hospitalization, use of anticonvulsants, inotropes, and fluid resuscitation, death before hospital discharge, length of hospital stay, and referral of patients to specialized follow-up.

**Funding:** GFTV and AN received funds from the Protecting Brain and Saving Futures (PBSF) Organization (www.pbsf.com.br). Both authors were involved in the design of the study, data collection and analysis, writing of the final manuscript, and decision to publish. PBSF Organization is registered with CNPJ 28.905.143/0001-61, headquartered at Av. Nove de Julho, 3229 – 13rd floor, room 1301 – Jardim Paulista, São Paulo, SP, Brazil – 01407-000, E-mail: contato@pbsf.com.br; Phone: + 55 (11) 3044-2557.

**Competing interests:** Gabriel FT Variane and Alexandre Netto are the owners of PBSF and Guilherme M Sant'Anna has no conflicts of interest and no financial relationships relevant to this article to disclose. All other authors work at PBSF.

**Abbreviations:** aEEG, Amplitude Integrated Electroencephalography; CBF, Cerebral Blood Flow; CHD, Congenital Heart Defect; cMRI, Cranial Magnetic Resonance Imaging; CNS, Central Nervous System; CSI, Central of Surveillance and Intelligence; cUS, Cranial Ultrasonography; ECMO, Extracorporeal Membrane Oxygenation; EEG, Electroencephalography; HIE, Hypoxic-Ischemic Encephalopathy; HR, Heart Rate; hsPDA, Hemodynamically Significant Patent Ductus Arteriosus; ICU, Intensive Care Units; IVH, Intraventricular Hemorrhage; MLPT, Moderate and Late Preterm Infants; NIBP, Non-invasive Blood Pressure; NICU, Neonatal Intensive Care Unit; NIRS, Near Infrared Spectroscopy; PBSF, Protecting Brains & Saving Futures; $pCO_2$, Partial Pressure of Carbon Dioxide; PGE, Prostaglandin; RCT, Randomized Controlled Trials; $rScO_2$, Cerebral Tissue Oxygenation; $rSO_2$, Cerebral and Somatic Tissue Oxygenation; $rSrO_2$, Renal Tissue Oxygenation; $SpO_2$, Pulse Oximetry; SWC, Sleep-Wake Cycle; TH, Therapeutic Hypothermia; WHO, World Health Organization; WMI, White Matter Injury.

## Discussion

The study will provide evaluation of *PBSF Guidelines* adherence and its impact on clinical outcomes. Thus, data from this large prospective, multicenter, and observational study will help determine whether neonatal neurocritical care via telemedicine can be effective. Ultimately, it may offer the necessary framework for larger scale implementation and development of research projects using remote neuromonitoring.

## Trial registration

NCT03786497, *Registered 26 December 2018*, https://www.clinicaltrials.gov/ct2/show/NCT03786497?term=protecting+brains+and+saving+futures&draw=2&rank=1.

## Introduction

Neonates requiring intensive care are exposed to a variety of interventions and treatments that improves survival but can affect their neurological system. Epidemiological studies indicate that every year 1.15 million infants suffer a hypoxic-ischemic event at birth and 13 million are born prematurely [1–5]. Of these, a substantial number will develop moderate or severe neurological impairment. Furthermore, other disorders such as congenital heart disease, cerebral malformations, congenital and nosocomial infections, intraventricular hemorrhages, inborn errors of metabolism, seizures, and severe hemodynamic and ventilatory instability can compromise brain integrity and function [6–8]. Modern neonatal intensive care units (NICU) can provide strategies to prevent neurological impairment using standardized clinical protocols and training, neuromonitoring, and neuroimaging; the so called neonatal neurocritical care units or neuroNICUs.

Amplitude Integrated Electroencephalography (aEEG) is a non-invasive methodology for continuous monitoring of brain function at the bedside, specifically for determination of electrical background activity and sleep-wake cycle (SWC) patterns, and detection of seizures [9, 10]. Indeed, the use of a two-channel aEEG with raw EEG readings (aEEG/EEG) has showed improved accuracy for the detection of seizures on real-time [11, 12], an important effect as seizure burden is an independent risk factor for neurodevelopmental delay [13–16] and treatment of subclinical seizures is associated with improved neurodevelopmental outcomes [17]. Moreover, in infants with hypoxic ischemic encephalopathy (HIE), aEEG findings have good predictive value for neurological short- and long-term outcomes [18–21].

Another available methodology for brain monitoring at the bedside is the Near Infrared Spectroscopy (NIRS), a non-invasive tool that provides continuous measurements of cerebral tissue oxygenation and perfusion (rSO2). NIRS evaluates the balance between tissue oxygen delivery and consumption allowing the early identification of perfusion abnormalities and hemodynamic changes [22, 23]. Therefore, NIRS monitoring is important in several clinical scenarios including HIE, prematurity, shock, anemia, congenital heart disease and other cases that lead to hemodynamic and ventilatory instability [24–26].

According to the World Health Organization (WHO), telemedicine is defined as the delivery of health care services, where distance is a critical factor, by all health care professionals using information and communication technologies for the exchange of valid information for diagnosis, treatment and prevention of disease and injuries, research and evaluation, and for the continuing education of health care providers, all in the interest of advancing the health of

individuals and their communities [25]. WHO also listed four important elements to telemedicine: provide clinical support; intention to overcome geographic barriers; use of several types of information and communication technologies; and improve health outcomes [26]. This model portrays the natural evolution of healthcare in the digital era, resulting in reduction of the distance between two or more locations, promoting access to specific methodologies in addition to reducing the cost of structure. It can also work through live interactive audiovisual links, live video transmission or online viewing of stored education material [27–29].

Brazil is the largest South America country with approximately 3 million live births per year and a significant proportion of infants at high-risk for brain injury [30]. Like any middle-income country, the available resources for neonatal care are unequal. Therefore, telemedicine can be used as a tool for education, consultation, patient care and research. This model allows low-cost access of remote and underserved areas to specific methodologies and expertise [27–29]. Considering that less than 5% of Brazilian NICUs provide well-structured neurocritical care [31], and the implementation of that requires specialized resources, adequate equipment, and qualified personnel, the use of an advanced and protocolized telemedicine model may be a better solution to decrease these disparities. For that, in order to provide neurocritical care to a broader range of Brazilian NICUs, a protocolized telemedicine system called *Protecting Brains and Saving Futures (PBSF) Guidelines* was developed and implemented in some NICUs. The goals of the *PBSF Guidelines* are to establish remote neurocritical care units by providing adequate equipment, specialized training, implementation of standardized protocols, continuous neuromonitoring and expert support, quickly and at low cost.

The aim of this prospective multicenter and observational study is to describe the centers' adherence to the *PBSF Guidelines* and clinical outcomes over a 3-years period. We hypothesize that the adherence to the *PBSF Guidelines* will increase over time and this will be associated with improvement of clinical outcomes. Results of this study may provide the necessary background information on the use of telemedicine aiming to protect neonatal brains and save futures, and for the development of larger studies and initiatives.

## Materials and methods

### Study design and setting

This prospective, multicenter, and observational study will be performed in the 20 Brazilian NICU's located at 4 different states, of which 15 are private and 5 are part of the Brazilian free health care system or phylantropic. All units have adopted the *PBSF Guidelines* as a standard-of-care for high-risk newborns and agreed to participate in the study. Data will be collected for a period of 3 years, from Dec 2021 to Nov 2024.

### Ethics approval and consent to participate

The study followed the precepts of good clinical practice and has been performed in accordance with the Declaration of Helsinki. The study proposal was approved by the Research Ethics Committee of the Irmandade da Santa Casa de Misericórdia de São Paulo under the following approval numbers and dates: 3.142.318 on February 12th, 2019 (S1 and S2 Files); 3.357.239 on May 30th, 2019 (S3 and S4 Files); and 3.506.106 on August 13th, 2019 (S5 and S6 Files). In addition, it received formal authorization from the Clinical and Administrative Board of each center. The original protocol approved by ethics is available at S7 File. The written informed consent from parents or guardians will be obtained by trained research investigators prior to inclusion in the study. The same care under the *PBSF Guidelines* will be provided to patients who decline study entry. All data will be treated anonymously and confidentially, and in no phase of the study any name, image, or data that allows participants identification

| | STUDY PERIOD | | | | | |
|---|---|---|---|---|---|---|
| | Enrolment | Allocation | Post-allocation | | | Close-out |
| TIMEPOINT | -$t_1$ | 0 | $t_1$ | $t_2$ | $t_3$ | $t_4$ |
| **ENROLMENT:** | | | | | | |
| **Eligibility screen** | X | | | | | |
| **Informed consent** | X | | | | | |
| **Allocation** | | X | | | | |
| **INTERVENTIONS** | | | | | | |
| **Not applicable** | | | | | | |
| **ASSESSMENTS:** | | | | | | |
| Population demographics variables | X | X | | | | |
| Primary diagnosis variables | | X | | | | |
| Primary outcomes variables | | | ◄————————► | | | X |
| Secondary outcomes variables | | | ◄————————► | | | X |
| **DATA ANALYSIS:** | | | | | | |
| Descriptive analysis | | | | | X | X |
| Statistical analysis | | | | | X | X |

**Fig 1. Schedule of enrollment, interventions, and assessments.**

will be disclosed. The study was registered in ClinicalTrials.gov (NCT03786497). It is reported using the Standard Protocol Items Recommendations for Interventional Trials (SPIRIT). The SPIRIT schedule is showed in Fig 1 and the completed SPIRIT checklist is available in S1 Appendix. Any proposed amendments will be discussed with the Research Ethics Committee and each institution's Clinical and Administrative Board and communicated with all investigators, sponsor, trial participants and trial registry (Clinicaltrials.gov).

## Eligibility criteria

In this study, the NICUs eligibility criteria are to have adopted the PBSF guidelines and agree to participate in the study (S2 Appendix). The characteristics of each NICU will be recorded and described. In those NICUs, all infants admitted from birth up to 3 months of age, and submitted to the *PBSF Guidelines* during the study period will be eligible. Indications for the use of the *PBSF Guidelines* are provided on Box 1. The *PBSF Guidelines* is the standard-of-care for high-risk newborns at all these hospitals. Patients with genetic syndromes or malformation incompatible with life will not be included. Details on the *PBSF Guidelines* are provided in (S3 Appendix). Briefly, it includes provision of equipment and resources, connection between the associated NICU and the remote monitoring center called the *Central of Surveillance and Intelligence (CSI)*, training and teaching of all health care

Box 1. Indications for use of the *PBSF Guidelines* in the participating centers.

**Indications**

1. Extreme prematurity

2. Peri-intraventricular Hemorrhage

3. Hypoxic-ischemic encephalopathy (mild, moderate, or severe)

4. Congenital heart disease

5. Neonatal stroke

6. Congenital infections

7. Nosocomial infections

8. Inborn errors of metabolism

9. Severe hemodynamic/ventilatory instability

10. Seizures

11. Brain malformations

12. Central nervous system infection

13. Extracorporeal membrane oxygenationPBSF, protecting brains and saving futures.

professionals of each NICU, and customized multiparametric recordings of biological signals from each patient. At the CSI, a team is available 24 hours a day all year around, allowing for case discussions and simplified reports with brain monitoring information at the patient monitor's display every 6 hours.

## Variables

Population demographics: gender, type of delivery, gestational age at birth, gestational age $\leq$ 32 weeks and $<$ 37 weeks, Apgar score at 1 min, 5 min and 10 min, birth weight, current weight, inborn or out born, use of antenatal steroids and magnesium sulfate. Primary diagnosis: seizure, mild HIE, moderate or severe HIE/cooling, neurologic abnormalities or congenital central nervous system (CNS) anomalies, grade III or IV intraventricular hemorrhage (IVH) or hydrocephalus, periventricular leukomalacia, meningitis, neural tube defects, stroke, cyanotic congenital heart defect (CHD), prematurity, gestational age $\leq$ 32 weeks, meconium aspiration syndrome, cardiorespiratory instability (anemia, shock and respiratory alkalosis/acidosis), necrotizing enterocolitis, metabolic disease, extracorporeal membrane oxygenation (ECMO)/pre-ECMO. For descriptive analysis patients will be divided into two groups accordingly to their primary diagnosis: neurological or clinical. A minimal anonymized data set necessary to replicate future study findings is provided as (S4 Appendix).

## Outcomes

The primary outcomes are detailed on Box 2. Results will be described as n (%) or median [IQR] for the overall population and for each center. Then, we will compare changes over time for the 3 years of data collection within and between centers. Finally, those results will be compared between 2 major groups: neurologic vs clinical patients.

Secondary outcomes are outlined on Box 3. To address potential source of bias related to financial availability, the outcomes of interest will be also adjusted based on the unit primary profile: private, philanthropic, foundation or public.

## Sample size

Based on our current experience, an average of 5 patients per month with criteria for neurocritical care are admitted on each center, making a total of 3,240 eligible patients during the 3-year period. Considering a 30% loss due to refusal to consent, non-availability of the research team or missing data, we expect to recruit 2,268 infants from all study centers (around 756 patients per year).

---

### Box 2. Primary outcomes.

**Adherence to the PBSF Guidelines**

1. Use and duration of aEEG/EEG/NIRS monitoring

2. Use of neuroimaging exams

3. Number of Sub-specialties consultation (neurology and neurosurgery)

4. Number of primary neurologic or medical patients with aEEG, EEG and/or NIRS monitoring and the duration of the monitoring (hours)

5. Number of primary neurologic or medical patients with brain MRI, neurology consult, and neurosurgery consult

6. Number of clinical case discussions and videoconference meetings

**Clinical outcomes**

1. Number of electroencephalographic seizures during hospitalization

2. Use and types of anticonvulsants administered

3. Use and types of inotropes and fluid resuscitation administered during NICU stay

4. Death before hospital discharge

5. Length of hospital stay

6. Number of patients referred to high-risk infant follow-up*PBSF, protecting brains and saving futures. *Every attempt will be made to obtain 18–24 months follow-up of these infants, with a Bayley III evaluation.

---

Box 3. Secondary outcomes.

**Descriptive usage of PBSF Guidelines**

1. Number of remote communications between CSI and local team

2. Number of reports issued for aEEG / EEG exams with or without the use of NIRS

3. Number of patients who performed Therapeutic Hypothermia

4. Adverse effects of brain monitoring expressed by skin lesion due to electrode / sensor positioningPBSF, protecting brains and saving futures; CSI, Central of Surveillance and Intelligence; NIRS, near infrared spectroscopy; aEEG, amplitude integrated electroencephalography; EEG, electroencephalography; MRI, magnetic resonance image; US, ultrasonography.

## Data collection and management

Patient demographics, diagnosis and clinical outcomes will be extracted from the medical charts of each patient and entered in the PBSF database. The *adherence to the PBSF Guidelines* will be measured with the data collected by the *CSI* center. Details of these recordings are provided on S3 Appendix.

A flow chart of all eligible and enrolled patients will be produced, and the total number of patients will be reported, overall and per center.

## Statistical methods

Categorical variables will be analyzed by descriptive statistics and presented as number of valid cases and percentage (%). Numeric variables will be analyzed as mean and standard deviation, median and interquartile ranges, or confident intervals. Since, it is possible that the patient population is quite different across the sites, analyses for the overall population will account for site differences or for the possibility of correlated observations within a site.

To compare adherence and outcomes over time (i.e., between years 1, 2 and 3) the following statistical approach will be applied: a) a test for normality in each category using the Shapiro-Wilks (<50 observations) or Kolmogorov–Smirnov, b) if normality assumption holds, a Welch's F test will be done, c) if normality assumption still holds the ANOVA with post hoc Bonferroni will be performed. If the normality assumption is violated, a non-parametric test (Kruskal–Wallis) will be used to analyze categorical variables and for significant differences between the two groups (neurological vs clinical). A final $p$ value of <0.05 will be considered statistically significant. Analysis of variability (distribution) per site for each variable of the 2 main outcomes will also be performed and Run Chart will be generated to follow the evolution of the outcomes over time for the 2 groups of patients: neurologic and clinical. All analysis will be done using the StataCorp. 2021. Stata Statistical Software: Release 17. College Station, TX: StataCorp LLC.

## Missing data

As missing data can impact study validity by reducing sample size and statistical power of subgroup analyses, a comprehensive strategy was developed. First, missing data will be defined as

values that are not available and that would be meaningful for analysis if they were observed. Second, the proportion of missing data for each variable will be presented. Furthermore, analysis to assess if patients with missing data differ from those with complete information for a given variable will be performed. Finally, information on why data are missing will be collected.

All efforts will be made to prevent missing data by a) collecting only critical data elements, b) focus on routinely collected data in real-world practice, c) develop completion guidelines, and d) perform pilot-test forms for the first 10 patients. Also, an ongoing data review in the CSI will be conducted for data completeness and quality. Moreover, investigators of each site will be trained on data collection and the importance of core data.

For analysis, we will understand the missing data, select the appropriate methodology (ies) for handling it and examine the sensitivity of different approaches. Thus, for each core variable we will generate frequency distributions and graphical displays. Two major methods will be used during analysis to handle missing data. For the primary analysis we are planning to perform a complete case analysis (use only cases with complete data) and them apply likelihood methods, i.e. mixed effect models to find the values of parameters which maximize the known likelihood value.

## Discussion

This large prospective, multicenter, and observational study will describe the use and adherence to a unique telemedicine neonatal neurocritical care guidelines and clinical outcomes. As a multidisciplinary approach to care for newborns with high-risk of brain injury has increased over the last 10 years, the implementation of an organized, low-cost, and high-expertise telemedicine guidelines has the potential to streamline clinical care improving outcomes.

The *PBSF Guidelines* defines high-risk neonates, with clear indications for neonatal neurocritical care. Among the etiologies associated with impaired neurodevelopmental outcomes, neonatal asphyxia represents the third cause of neonatal death worldwide (23%) [2, 3, 32–34]. In newborns with moderate or severe HIE the risk of death, cerebral palsy and/or disability among survivors is high. In addition, some disability might also occur in mild HIE infants [4, 6, 7, 35]. Neonatal seizures have been reported in approximately 3.5 per 1000 live births of term neonates [17] and several conditions are associated with increased risks for developing seizures such as perinatal asphyxia and ischemic stroke. Indeed, it is estimated that perinatal asphyxia is the leading cause of seizures in the neonatal period, accounting for 40 to 60% of all seizures in term newborns [36, 37]. Seizure burden is associated with more severe brain injury and worse neurodevelopmental outcomes after adjusting for underlying severity of brain injury. Status epilepticus is also associated with post neonatal epilepsy [17]. Since seizures are difficult to diagnose clinically, continuous monitoring by using aEEG, EEG or video aEEG/EEG will help the identification of sub-clinical events allowing immediate recognition and treatment, which can reduce seizure burden and improve neurodevelopmental outcomes [13–16].

Preterm infants are also at high risk for neurodevelopmental problems, and different areas may be affected such as neurosensory, cognitive, and behavioral functioning. The more immature, the greater the risks [8, 38] and extreme prematurity has been associated with moderate or severe neurocognitive deficits during childhood [39]. Moderate and late preterm infants (MLPT) can also have worse cognitive, language, motor development, social-emotional competence at 2 years of age when compared to term infants [40]. Other disorders such as cerebral malformations, infections, inborn errors of metabolism, and severe hemodynamic and ventilatory instability also carry risks of brain injury and seizures. Moreover, CHD affects from 4 to

75 per 1,000 live births and has been recognized as high risk for neurodevelopmental problems [41, 42]. In CHD, infants brain injury may follow episodes of cerebral hypoxia or hypoperfusion that can occur during the pre-, intra- and/or postoperative periods, in addition to biological and environmental factors [43–45]. Indeed, more than 50% of CHD patients have been reported with neurodevelopmental deficiencies [44–47] that may persist into adolescence [47]. All these situations are included as indications for the use of the *PBSF Guidelines*.

In recent decades, brain monitoring has become increasingly common in NICUs [48] but not yet a consolidated practice in all services, especially in low- and middle- income countries (LMIC), where the lack of structure, resources, and professional training is a major problem. Brazil is a large country with a population of > 200 million people and approximately 3 million births per year, with great financial disparity. This discrepancy directly affects the Brazilian health system, with differences in availability of financial, material, and personal resources, including adequate training of professionals. A prospective cross-sectional national survey conducted by our group evaluated the assessment of HIE and TH practices in Brazil. A total of 1,092 professionals answered and 62% reported using TH in their units. However only 13% received specific training for HIE assessment, 22% did not use any neurologic score, and only 12% had brain monitoring readily available, showing the need for well-established neurocritical care guidelines [31].

There are a variety of clinical applications of video aEEG/EEG monitoring and NIRS in neonates at high-risk for brain injury:

1. *Seizures*: clinical signs of seizures are easy to evaluate and overdiagnosis occurs frequently, resulting in unnecessary treatment [21, 33, 36]. Neonatal seizures are frequently underdiagnosed since more than 80–90% of seizures are subclinical, resulting in undertreatment. Indeed, the accuracy for seizure detection with clinical evaluation alone is < 9% [37]. New devices associate aEEG with raw EEG readings and even video imaging (video aEEG/EEG), providing higher accuracy for the detection of seizures [12, 36, 48, 49]. Shah et al. evaluated term newborns with clinical seizures monitored with aEEG with simultaneous continuous conventional EEG for seizure detection. In 41 non-status epilepticus seizures, 31 were correctly identified (sensitivity, 76%; specificity, 78%; PPV, 78%; NPV, 78%). However, they found a lower sensitivity (27–56%) when aEEG was assessed without the conventional EEG findings [11].

2. *Hypoxic Ischemic Encephalopathy*: Three of the six major clinical trials of TH used an abnormal aEEG background activity as entry criteria [50–52]. Indeed, an abnormal aEEG may improve selection of neonates to be treated [53]. However, the presence of a normal early aEEG should not exclude indication of TH in infants with neurological evidence of moderate to severe HIE [54]. After the implementation of TH, the predictive value of aEEG has been evaluated. Chandrasekaran et al. found that persistently abnormal trace on aEEG at ≥ 48 hours of life was associated with adverse long-term outcomes and a systematic review [55] found that maximum predictive reliability was achieved at 72 h of life with a post-test probability of 95.7% (95% CI: 84.4 to 98.5%). Therefore, aEEG is a valuable tool in infants with HIE to be used for the classification of the level of encephalopathy and as a predictor of long-term outcomes. The persistence of elevated values of rScO2 was also associated with abnormal neuroimaging and adverse neurodevelopmental outcomes as low cerebral metabolism indicates low oxygen use, cerebral hyperperfusion and impaired autorregulation [56–58].

3. *Preterm infants*: the incidence of seizures in preterm infants has been reported between 4 to 48%, and seizures were associated with adverse outcomes including IVH, white matter

injury (WMI), cognitive impairment and neonatal death [13, 16, 59, 60]. The presence of SWC and continuous background pattern in the first week of life were associated to good neurodevelopmental outcomes, while absence of SWC and abnormal background activity were associated with neonatal death, severe IVH, WMI and neurological impairment at 2 years of age [61–64]. In extremely premature newborns during the first days of life, the occurrence of episodes of hypoxia/hyperoxia, hemodynamic and ventilatory instability, and systemic hypotension with decreased cerebral flow are common. For that, NIRS monitoring has been used extensively used in this population. A phase II randomized controlled trial conducted by the SafeBoosC Consortium evaluated newborns $\leq$ 72 h of life to demonstrate the feasibility and effectiveness of implementing continuous NIRS monitoring with a clinical management protocol. Neonates monitored with NIRS had lower burden of cerebral hypoxia or hyperoxia compared to controls and a trend towards lower IVH and mortality rates [65]. A hemodynamically significant patent ductus arteriosus (hsPDA) may be associated with increased pulmonary blood flow and decreased systemic blood flow. Chock et. al found that low renal oxygenation values (rSrO2), with rSrO2 < 66%, were associated with the presence of hemodynamically significant PDA [24]. Other studies have successfully demonstrated association between PDA closure and significant increases on cerebral and renal saturation [66, 67].

4. *Other Applications*: ventilatory instability (respiratory acidosis/alkalosis), metabolic disorders, inborn errors of metabolism, cerebral malformations, stroke, congenital infections, sepsis, meningitis and need of extracorporeal circulation (ECMO) [48, 68, 69]. In 20% of infants with CHD electrographic seizures were detected, including repetitive seizures and status epilepticus [70]. Early recovery to normal aEEG background activity and presence of SWC was associated with good neurological outcomes after cardiac surgery [46, 71]. Perioperative electrical seizures were found in 30% of infants, of which only ¼ showed any clinical correlation. Failure to recover the background activity to a continuous pattern $\geq$ 48 h postoperatively was associated with neurodevelopmental delay. The persistence of abnormal aEEG for seven days after surgery was also highly associated with death [46].
NIRS monitoring may help to evaluate efficacy or need for additional interventions such as ventilator changes, use of diuretics, change in prostaglandin (PGE) dose, need for blood transfusion or early surgical intervention. A rSrO2 < 30% was correlated with increased Intensive Care Units (ICU) stay, and rScO2 and rSrO2 < 45% and < 40% respectively, were correlated with need of ECMO and death [72]. NIRS monitoring was also useful to diagnose early renal insufficiency after surgical intervention [73–75].

5. *Anemia and shock*: tissue oxygen delivery, somatic and cerebral saturations are directly affected by hemoglobin levels, especially in scenarios of important anemia, suggesting that NIRS monitoring may play a role in the indication of blood transfusion. A significant increase in rScO2 values was reported in children with rScO2 <55% treated with blood transfusion [76]. NIRS has also been demonstrated as a good early indicator of abnormal somatic perfusion and shock in different populations including newborns [77–79]. In the setting of shock, two site NIRS monitoring (cerebral and somatic) may be particularly useful in cases where autoregulation remains intact and somatic tissue oxygenation may be compromised earlier [80].

Implementation of a NeuroNICU requires specialized education, training, use of validated protocols and implementation of specific methodologies. Financial resources are needed for equipment, such as servo-controlled temperature and brain monitoring devices. Lack of specialized assistance and resources may result in inadequate interventions with possible lack of

benefit or even harm to the patients. A systematic review found no reduction in mortality with the use of TH in LMIC countries [81] and possible reasons to justify this finding include inefficiency of the low technology cooling devices and lack of optimal care. Brain monitoring using conventional EEG requires skilled technologists, experienced neurologist interpretation which may not be readily available in many centers, especially for continuous bedside monitoring. aEEG and NIRS monitoring can be interpreted at the bedside but also requires training and experienced users [82, 83]. Unfortunately, the majority of neonatal units lack adequate longitudinal training, specialized staff, and access to all these resources, especially in LMIC. This is what motivated our group to consider telemedicine as a solution in this scenario.

Differently from the previous studies, the *PBSF Guidelines* (S3 Appendix) is unique for the use of an advanced telemedicine system to provide neurocritical care. The guidelines include continuous neuromonitoring with data recording and real time feedback from experts, provision of equipment and resources to the NICUs, and short- and long-term teaching to the professionals working at the units (S2 Appendix). Telemedicine has several applications including education, consultation, practice, and research [28]. This type of communication has shown results in continuing education among physicians and nurses, and has assisted healthcare professionals to improve health disparities, promoting access to specialized care [27, 29]. Live interactive audiovisual materials, such as live streaming video and viewing stored educational material, are important to deliver tele-education; it promotes opportunities for continuous learning and development by dissemination of information and training health-care professionals [26, 28]. Telemedicine may also be helpful in case discussions for decision making which is useful to promote interaction between physicians, nurses, and patients. Udeh et al. exemplified some benefits of telemedicine including [84]: a) promote evidence-based best practices through checklists and prompting; b) enhance monitoring, early identification, and treatment of critical illness; c) improve coordination of care; and d) increase night-time vigilance. The use of telemedicine in ICUs is associated with decreased mortality and length of stay [84, 85]. Surprisingly, telemedicine has not been used on a significant scale worldwide, although this methodology is low-cost, feasible, clinically useful, and sustainable. It could be even more useful in LMIC countries due to the limited structure and resources.

Brazil allows several opportunities for telemedicine implementation, expanding the actions of healthcare professionals and integrating healthcare services [86]. We developed the *PBSF Guidelines* with a surveillance and intelligence center to support clinical decisions across the country. Telemedicine has been used by our group in the past few years to promote educational training and remote neuromonitoring but its impact on clinical outcomes needs to be formally evaluated. Thus, results of this 3 years prospective, multicenter, and observational study will help to determine the incidence of neonatal problems associated with brain injury and disabilities, the incidence of clinical and subclinical seizures, detection of brain hemodynamic/oxygenation instability and associated interventions. The study will be limited to the 20 NICUs using the *PBSF Guidelines* and variability in clinical practices, experience of health care providers and staffing of the units, will be analyzed as it may affect some of the outcomes. Enrollment of such large number of patients will require a strong coordination between the central unit and all 20 centers, which has already been established by the CSI. Practical or operational issues will be to assure that all eligible patients are identified in a daily basis and approached for consent. Data collection requires careful training of data extractors and timely and precise data entry. A computerized and safe double-checking system has been developed. Biological signals will be continuously collected and stored at the CSI and multimodal signal analysis has been developed. Although low, there are costs for the implementation and use of the *PBSF Guidelines*. Thus, an adjusted analysis based on the profile of the units will be

performed as costs might represent a limitation on the generalizability of the study findings. A future cost/benefit analysis is under planning.

## Supporting information

**S1 Appendix. SPIRIT 2013 checklist.**
(PDF)

**S2 Appendix. Characteristics of participating centers.**
(PDF)

**S3 Appendix. The Protecting Brains and Saving Futures (PBSF) Guidelines.**
(PDF)

**S4 Appendix. Minimal anonymized data set.**
(PDF)

**S1 File. Original ethical approval CEP3142318.**
(PDF)

**S2 File. Ethical approval CEP3142318.**
(PDF)

**S3 File. Original ethical approval CEP3357329_E1.**
(PDF)

**S4 File. Ethical approval CEP3357329_E1.**
(PDF)

**S5 File. Original ethical approval CEP3506106_E2.**
(PDF)

**S6 File. Ethical approval CEP3506106_E2.**
(PDF)

**S7 File. Original protocol approved by ethics.**
(PDF)

## Author Contributions

**Conceptualization:** Gabriel Fernando Todeschi Variane, Maurício Magalhães, Alexandre Netto, Daniela Pereira Rodrigues, Renato Gasperini, Guilherme Mendes Sant'Anna.

**Data curation:** Gabriel Fernando Todeschi Variane, Maurício Magalhães, Rafaela Fabri Rodrigues Pietrobom, Alexandre Netto, Daniela Pereira Rodrigues, Guilherme Mendes Sant'Anna.

**Formal analysis:** Gabriel Fernando Todeschi Variane, Maurício Magalhães, Rafaela Fabri Rodrigues Pietrobom, Alexandre Netto, Daniela Pereira Rodrigues, Guilherme Mendes Sant'Anna.

**Funding acquisition:** Gabriel Fernando Todeschi Variane, Alexandre Netto.

**Investigation:** Gabriel Fernando Todeschi Variane, Rafaela Fabri Rodrigues Pietrobom, Daniela Pereira Rodrigues, Guilherme Mendes Sant'Anna.

**Methodology:** Gabriel Fernando Todeschi Variane, Rafaela Fabri Rodrigues Pietrobom, Daniela Pereira Rodrigues, Renato Gasperini, Guilherme Mendes Sant'Anna.

**Project administration:** Gabriel Fernando Todeschi Variane, Maurício Magalhães, Alexandre Netto, Guilherme Mendes Sant'Anna.

**Resources:** Gabriel Fernando Todeschi Variane.

**Supervision:** Maurício Magalhães, Guilherme Mendes Sant'Anna.

**Validation:** Gabriel Fernando Todeschi Variane, Maurício Magalhães, Rafaela Fabri Rodrigues Pietrobom, Alexandre Netto, Daniela Pereira Rodrigues, Guilherme Mendes Sant'Anna.

**Visualization:** Gabriel Fernando Todeschi Variane, Maurício Magalhães, Rafaela Fabri Rodrigues Pietrobom, Alexandre Netto, Daniela Pereira Rodrigues, Renato Gasperini, Guilherme Mendes Sant'Anna.

**Writing – original draft:** Gabriel Fernando Todeschi Variane, Maurício Magalhães, Rafaela Fabri Rodrigues Pietrobom, Alexandre Netto, Daniela Pereira Rodrigues, Renato Gasperini, Guilherme Mendes Sant'Anna.

**Writing – review & editing:** Gabriel Fernando Todeschi Variane, Maurício Magalhães, Rafaela Fabri Rodrigues Pietrobom, Alexandre Netto, Daniela Pereira Rodrigues, Renato Gasperini, Guilherme Mendes Sant'Anna.

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
