## [Decision Letter · Decision Letter 0]

18 Aug 2021

PONE-D-21-10350

Protecting Brains and Saving Futures Guidelines: a prospective, multicenter, and observational study on the use of telemedicine for neonatal neurocritical care in Brazil

PLOS ONE

Dear Dr. Variane,

Thank you for submitting your manuscript to PLOS ONE. After careful consideration, we feel that it has merit but does not fully meet PLOS ONE’s publication criteria as it currently stands. Therefore, we invite you to submit a revised version of the manuscript that addresses the points raised during the review process.

Please follow the reviewer's advice for further consideration.

We look forward to receiving your revised manuscript.

Kind regards,

Kazumichi Fujioka

Academic Editor

PLOS ONE

Journal Requirements:

3. Please provide additional details regarding participant consent. In the ethics statement in the Methods and online submission information, please ensure that you have specified what type you will obtain (for instance, written or verbal, and if verbal, how it was documented and witnessed).

4. We suggest you thoroughly copyedit your manuscript for language usage, spelling, and grammar. If you do not know anyone who can help you do this, you may wish to consider employing a professional scientific editing service. 

5. We noted that you submitted this Study Protocol as a clinical trial, but according to your description and the WHO definition of clinical trials we would not consider this a clinical trial but rather an observational study.

6. We note that the grant information you provided in the ‘Funding Information’ and ‘Financial Disclosure’ sections do not match. 

7. Thank you for stating the following in the Competing Interests section: 

"Gabriel FT Variane and Alexandre Netto are the owners of PBSF. Guilherme M Sant’Anna has no conflicts of interest and no financial relationships relevant to this article to disclose. All other authors work at PBSF." 

8. We note that you have indicated that data from this study are available upon request. PLOS only allows data to be available upon request if there are legal or ethical restrictions on sharing data publicly. For more information on unacceptable data access restrictions, please see http://journals.plos.org/plosone/s/data-availability#loc-unacceptable-data-access-restrictions. 

9. Your ethics statement should only appear in the Methods section of your manuscript. If your ethics statement is written in any section besides the Methods, please delete it from any other section. 

10. Please include captions for your Supporting Information files at the end of your manuscript, and update any in-text citations to match accordingly. Please see our Supporting Information guidelines for more information: http://journals.plos.org/plosone/s/supporting-information. 

Reviewers' comments:

Reviewer's Responses to Questions

**Comments to the Author**

1. Is the manuscript technically sound, and do the data support the conclusions?

Reviewer #1: Yes

Reviewer #2: Partly

2. Has the statistical analysis been performed appropriately and rigorously? 

Reviewer #1: N/A

Reviewer #2: N/A

3. Have the authors made all data underlying the findings in their manuscript fully available?

Reviewer #1: Yes

Reviewer #2: No

4. Is the manuscript presented in an intelligible fashion and written in standard English?

Reviewer #1: Yes

Reviewer #2: Yes

5. Review Comments to the Author

Reviewer #1: The authors present the protocol for an upcoming study that will evaluate the adherence to proposed guidelines that utilize telemedicine for neonatal neurocritical care in Brazil and clinical outcomes for infants in the NICU enrolled in the study over a 3-year period. The authors provide background and rationale for the study, details about the Guidelines, key outcomes of interest, as well as plans for statistical analyses. The manuscript will be strengthened if the authors consider the following points.

1. There are some missing words, incorrect word usage, and instances of awkward phrasing. For example, line 48, "neonatal neurocritical via" should be "neonatal neurocritical care via", line 61 "varied" should be "variety", lines 62-64, the text switches between present and past tense, line 120 "background information the use" should be "background information on the use", in Box 2, #4 under Adherence, "Number primary" should be "Number of primary", line 252 "are associate" should be "are associated", line 275 "as indication" should be "as indications", line 376 "includes" should be "include", and line 400 "these 3 years" should be "this 3-year".

2. In the Statistical Methods section, authors can provide more details. For example, there are 20 sites being included in the study. It is possible that the patient population is quite different across the sites, yet the proposed analyses do not account for site differences or for the possibility of correlated observations within a site since doctors/nurses at one site will likely care for the infants within that site similarly. Authors are studying a lot of primary and secondary outcomes, yet they state that p<0.05 will be considered statistically significant - authors should consider a form of multiple comparison adjustment (beyond the post-hoc pairwise comparisons adjusted using the Bonferroni correction after the ANOVA). Authors plan to use ANOVA for comparing outcomes over time, but have authors considered alternative approaches if the assumptions of the ANOVA are not met? What software will be used for the analyses? Are there any plans for handling missing data? In line 229, do authors mean "confidence intervals" instead of "coefficient intervals"?. The reason for the planned use of Kaplan Meier curves is not entirely clear (lines 235-236), as it is not clear what the time scale is. Generally, in Kaplan-Meier curves, one has everyone that is potentially at risk for an outcome at the start of the study and then follows individuals until events occur. Will the time scale be time since birth?

Reviewer #2: Congratulations to the authors for this prospective, multicenter, and observational study that will be conducted in 20 Brazilian NICUs using pre-specified PBSF guidelines related to neonatal neurocritical care via telemedicine.

Background mentions the importance of EEG, amplitude integrated EEG, near infrared spectroscopy, and neuroimaging, besides the use of telemedicine in the neonatal neurocritical care, and the justification of the study.

The primary goal is to describe adherence to the PBSF guidelines and the clinical outcomes in patients until 3 months old, by NICU and over a 3-year period.

Methods mention eligibility patient criteria, data collection and management, sample size, and statistical analyses; and detailed appendixes.

Authors need to explain the eligibility criteria regarding the 20 NICU enrolled in the study.

Discussion justifies the study population – high-risk neonates with indications for neonatal neurocritical care; and it refers that the results may help to implement the guidelines in a large scale and the development of research projects using remote neuromonitoring.

Please use the space provided to explain your answers to the questions.

1. Is the manuscript technically sound, and do the data support the conclusions? This does not apply yet because it is a study protocol.

2. Has the statistical analysis been performed appropriately and rigorously? Not applicable yet because it is a study protocol.

3. Have the authors made all data underlying the findings in their manuscript fully available? No; this is a study protocol without results yet.

4. Is the manuscript presented in an intelligible fashion and written in standard English? Yes

6. PLOS authors have the option to publish the peer review history of their article (what does this mean?). If published, this will include your full peer review and any attached files.

Reviewer #1: No

Reviewer #2: No

---

## [Author Response · Author response to Decision Letter 0]

16 Nov 2021

Response to Reviewers 

PONE-D-21-10350: Protecting Brains and Saving Futures Guidelines: a prospective, multicenter, and observational study on the use of telemedicine for neonatal neurocritical care in Brazil.

Editor comments: 

R: Dr Guilherme Sant’Anna, Full Professor of Pediatrics, McGill University Health Center, Montreal, Canada, 

R: A copy of the manuscript with highlighted changes was uploaded.

R: A clean copy of the edit manuscript was uploaded.

4. We noted that you submitted this Study Protocol as a clinical trial, but according to your description and the WHO definition of clinical trials we would not consider this a clinical trial but rather an observational study.

R: The manuscript was now submitted as an observational study. 

5. We note that the grant information you provided in the ‘Funding Information’ and ‘Financial Disclosure’ sections do not match. When you resubmit, please ensure that you provide the correct grant numbers for the awards you received for your study in the ‘Funding Information’ section.

R: No grant awards were received for this study. We provided the financial information under ‘Financial Disclosure’. 

6. Thank you for stating the following in the Competing Interests section: 

"Gabriel FT Variane and Alexandre Netto are the owners of PBSF. Guilherme M Sant’Anna has no conflicts of interest and no financial relationships relevant to this article to disclose. All other authors work at PBSF." 

R: We have included the requested sentence: "This does not alter our adherence to PLOS ONE policies on sharing data and materials.” This updated Competing Interest statement was included in the cover letter.

7. We note that you have indicated that data from this study are available upon request. PLOS only allows data to be available upon request if there are legal or ethical restrictions on sharing data publicly. 

In the revised cover letter, we added:

a) There are no data available currently as the manuscript is a protocol and not the study. However, there are no legal restrictions on sharing a de-identified data without potentially sensitive information.

b) A minimal anonymized data set necessary to replicate future study findings is provided as a Supporting Information file.

8. Your ethics statement should only appear in the Methods section of your manuscript. If your ethics statement is written in any section besides the Methods, please delete it from any other section. 

R: This was checked. 

R: This was included. 

Reviewers' comments:

Reviewer's Responses to Questions

Comments to the Author

1. Is the manuscript technically sound, and do the data support the conclusions?

Reviewer #1: Yes

Reviewer #2: Partly

R: We addressed Reviewer #2 comment below: see answer to questions 2 to 8. 

2. Has the statistical analysis been performed appropriately and rigorously?

Reviewer #1: N/A

Reviewer #2: N/A

3. Have the authors made all data underlying the findings in their manuscript fully available?

The PLOS Data policy requires authors to make all data underlying the findings described in their manuscript fully available without restriction, with rare exception (please refer to the Data Availability Statement in the manuscript PDF file). The data should be provided as part of the manuscript or its supporting information or deposited to a public repository. For example, in addition to summary statistics, the data points behind means, medians and variance measures should be available. If there are restrictions on publicly sharing data—e.g. participant privacy or use of data from a third party—those must be specified.

Reviewer #1: Yes

Reviewer #2: No

R: There is no data yet collected. Thus, there is no summary statistics, or data points behind means, medians and variance measures available. We fully agree with PLOS Data policy, and this will be provided when the study is done, and results analyzed. 

4. Is the manuscript presented in an intelligible fashion and written in standard English?

Reviewer #1: Yes

Reviewer #2: Yes

R: Thank you. 

5. Review Comments to the Author

Reviewer #1: The authors present the protocol for an upcoming study that will evaluate the adherence to proposed guidelines that utilize telemedicine for neonatal neurocritical care in Brazil and clinical outcomes for infants in the NICU enrolled in the study over a 3-year period. The authors provide background and rationale for the study, details about the Guidelines, key outcomes of interest, as well as plans for statistical analyses. The manuscript will be strengthened if the authors consider the following points.

1. There are some missing words, incorrect word usage, and instances of awkward phrasing. For example, line 48, "neonatal neurocritical via" should be "neonatal neurocritical care via", line 61 "varied" should be "variety", lines 62-64, the text switches between present and past tense, line 120 "background information the use" should be "background information on the use", in Box 2, #4 under Adherence, "Number primary" should be "Number of primary", line 252 "are associate" should be "are associated", line 275 "as indication" should be "as indications", line 376 "includes" should be "include", and line 400 "these 3 years" should be "this 3-year".

R: We did our best to correct all these mistakes (and others). The manuscript was extensively revised. 

2. In the Statistical Methods section, authors can provide more details. For example, there are 20 sites being included in the study. It is possible that the patient population is quite different across the sites, yet the proposed analyses do not account for site differences or for the possibility of correlated observations within a site since doctors/nurses at one site will likely care for the infants within that site similarly. 

R: Excellent point. Under the primary outcomes section, we outlined that “outcomes will be described for the overall population and for each center”. We now added in Statistical methods the following sentence: “Since, it is possible that the patient population is quite different across the sites, analyses for the overall population will account for site differences or for the possibility of correlated observations within a site”.

3. Authors are studying a lot of primary and secondary outcomes, yet they state that p<0.05 will be considered statistically significant - authors should consider a form of multiple comparison adjustment (beyond the post-hoc pairwise comparisons adjusted using the Bonferroni correction after the ANOVA). 

R: Very interesting comment. We reviewed this subject when preparing the protocol and found limited guidance as to which method(s) should be used to account for multiplicity during the statistical analysis. We found that the Bonferroni adjustment is one of the most used approaches (Bland JM, Altman DG. Multiple significance tests: the Bonferroni method. BMJ 1995;310:170), is a very stringent criterion, and compute the adjusted P values by directly multiplying the number of simultaneously tested hypotheses. Therefore, this method has been well acknowledged to be much conservative especially when there are many hypotheses being simultaneously tested and/or hypotheses are highly correlated as in the case of the proposed protocol. We would be grateful if the reviewer could make suggestions on different methods we could incorporate in the protocol analysis. 

4. Authors plan to use ANOVA for comparing outcomes over time, but have authors considered alternative approaches if the assumptions of the ANOVA are not met? 

R: Excellent point. Yes, we added this information on page 12. 

“To compare adherence and outcomes over time (i.e., between years 1, 2 and 3) the following statistical approach will be applied: a) a test for normality in each category using the Shapiro-Wilks (<50 observations) or Kolmogorov–Smirnov, b) if normality assumption holds, a Welch's F test will be done, c) if normality assumption still holds the ANOVA with post hoc Bonferroni will be performed. If the normality assumption is violated, a non-parametric test (Kruskal–Wallis) will be used to analyze categorical variables and for significant differences between the two groups (neurological vs clinical)”.

5. What software will be used for the analyses? 

R: StataCorp. 2021. Stata Statistical Software: Release 17. College Station, TX: StataCorp LLC.

6. Are there any plans for handling missing data? 

R: This is a very important issue that was not clearly addressed in the manuscript as it can impact study validity (reduce sample size and statistical power for subgroup analyses). Thank you for the question. 

First, missing data will be defined as values that are not available and that would be meaningful for analysis if they were observed (Little, et al., NEJM 2012). Second, we will present the proportion of missing data for each variable, analyse if patients with missing data differ from those with complete information for a given variable and if so, how? We will also collect information on why data are missing. 

All efforts will be made to prevent missing data by collecting only critical data elements, focus on routinely collected data in real-world practice, develop completion guidelines and perform pilot-test forms for the first 10 patients. We will conduct ongoing data review in the CSI for data completeness and quality and investigators of each site will be trained on data collection and the importance of core data. Also, we estimated a 30% loss due to refusal to consent, non-availability of the research team or missing data for the sample size to be enrolled. 

For analysis, we will understand the missing data, select the appropriate methodology (ies) for handling it and examine the sensitivity of different approaches. 

For each core variable we will generate frequency distributions and graphical displays. Two major methods will be used during analysis to handle missing data. For the primary analysis we are planning to perform a complete case analysis (use only cases with complete data) and them apply likelihood methods, i.e. mixed effect models to find the values of parameters which maximize the known likelihood value. 

This information was inserted on page 13. 

7. In line 229, do authors mean "confidence intervals" instead of "coefficient intervals"?

R: Thanks for pointing the mistake. This was corrected. 

8. The reason for the planned use of Kaplan Meier curves is not entirely clear (lines 235-236), as it is not clear what the time scale is. Generally, in Kaplan-Meier curves, one has everyone that is potentially at risk for an outcome at the start of the study and then follows individuals until events occur. Will the time scale be time since birth?

R: That’s a very good point again. We changed for Run Charts will be generated to follow the evolution of the outcomes over time...” This information was inserted in the manuscript, page 13. 

Reviewer #2: Congratulations to the authors for this prospective, multicenter, and observational study that will be conducted in 20 Brazilian NICUs using pre-specified PBSF guidelines related to neonatal neurocritical care via telemedicine. Background mentions the importance of EEG, amplitude integrated EEG, near infrared spectroscopy, and neuroimaging, besides the use of telemedicine in the neonatal neurocritical care, and the justification of the study.

1. The primary goal is to describe adherence to the PBSF guidelines and the clinical outcomes in patients until 3 months old, by NICU and over a 3-year period.

R: Thank you. 

2. Methods mention eligibility patient criteria, data collection and management, sample size, and statistical analyses; and detailed appendixes. Authors need to explain the eligibility criteria regarding the 20 NICU enrolled in the study.

R: That’s an excellent comment. The NICUs eligibility criteria are to have adopted the PBSF guidelines and agree to participate in the study. The characteristics of each NICU will be recorded and described. This information was inserted in the manuscript under eligibility criteria, page 8. 

3. Discussion justifies the study population – high-risk neonates with indications for neonatal neurocritical care; and it refers that the results may help to implement the guidelines in a large scale and the development of research projects using remote neuromonitoring.

R: Thank you. 

4. PLOS authors have the option to publish the peer review history of their article (what does this mean?). If published, this will include your full peer review and any attached files.

If you choose “no”, your identity will remain anonymous, but your review may still be made public.

R: We agree to have all history published.

---

## [Editor Report · Decision Letter 1]

30 Dec 2021

Protecting Brains and Saving Futures Guidelines: a prospective, multicenter, and observational study on the use of telemedicine for neonatal neurocritical care in Brazil

PONE-D-21-10350R1

Dear Dr. Variane,

We’re pleased to inform you that your manuscript has been judged scientifically suitable for publication and will be formally accepted for publication once it meets all outstanding technical requirements.

Kind regards,

Kazumichi Fujioka

Academic Editor

PLOS ONE
---

## [Editor Report · Acceptance letter]

4 Jan 2022

PONE-D-21-10350R1 

Protecting Brains and Saving Futures Guidelines: a prospective, multicenter, and observational study on the use of telemedicine for neonatal neurocritical care in Brazil 

Dear Dr. Variane:

I'm pleased to inform you that your manuscript has been deemed suitable for publication in PLOS ONE. Congratulations! Your manuscript is now with our production department. 

Kind regards, 

on behalf of

Dr. Kazumichi Fujioka 

Academic Editor

PLOS ONE